# Investigation on Atomic Bonding Structure of Solution-Processed Indium-Zinc-Oxide Semiconductors According to Doped Indium Content and Its Effects on the Transistor Performance

**DOI:** 10.3390/ma15196763

**Published:** 2022-09-29

**Authors:** Dongwook Kim, Hyeonju Lee, Bokyung Kim, Sungkeun Baang, Kadir Ejderha, Jin-Hyuk Bae, Jaehoon Park

**Affiliations:** 1Department of Electronic Engineering, Hallym University, Chuncheon 24252, Korea; 2Department of Physics, Faculty of Science and Arts, Bingol University, Bingol 12000, Turkey; 3School of Electronics Engineering, Kyungpook National University, Daegu 41566, Korea; 4School of Electronic and Electrical Engineering, Kyungpook National University, Daegu 41566, Korea

**Keywords:** solution-processed IZO semiconductor, thin-film transistor, atomic structure, doping

## Abstract

The atomic composition ratio of solution-processed oxide semiconductors is crucial in controlling the electrical performance of thin-film transistors (TFTs) because the crystallinity and defects of the random network structure of oxide semiconductors change critically with respect to the atomic composition ratio. Herein, the relationship between the film properties of nitrate precursor-based indium-zinc-oxide (IZO) semiconductors and electrical performance of solution-processed IZO TFTs with respect to the In molar ratio was investigated. The thickness, morphological characteristics, crystallinity, and depth profile of the IZO semiconductor film were measured to analyze the correlation between the structural properties of IZO film and electrical performances of the IZO TFT. In addition, the stoichiometric and electrical properties of the IZO semiconductor films were analyzed using film density, atomic composition profile, and Hall effect measurements. Based on the structural and stoichiometric results for the IZO semiconductor, the doping effect of the IZO film with respect to the In molar ratio was theoretically explained. The atomic bonding structure by the In doping in solution-processed IZO semiconductor and resulting increase in free carriers are discussed through a simple bonding model and band gap formation energy.

## 1. Introduction

Oxide semiconductor thin-film transistors (TFT) have recently garnered considerable attention as next-generation display materials owing to the remarkable development of flexible displays [1,2,3,4]. In particular, the possibility of solution-processing oxide semiconductor materials encourages ultra-low-cost mass production via the printing process. Over the past 20 years, oxide semiconductors have significantly improved in chemical and technical processes, and solution-processed TFTs with high mobilities of approximately 10 cm·V^−1^s^−1^ or more have been reported [5,6]. Particularly, solution-processed oxide semiconductor TFTs using nitrate precursors exhibit enhanced film quality because of the high volatility of the by-products in the reaction process [7,8,9].

With increasing interest in solution-processed oxide semiconductors, TFTs with high electrical mobilities of over 30 cm·V^−1^s^−1^ and low driving voltages of less than 2 V have been reported [10,11]. Moreover, studies on process developments such as atomic composition ratio, doping, and chemical treatment have remarkably improved TFT performance [12,13]. Among them, in the studies focusing on the atomic composition ratio, the field-effect mobilities in the range 0.5–10.0 cm·V^−1^s^−1^ were controlled by varying the ratio of In:Zn or In:Zn:Ga, and the thickness, roughness, and crystallinity of the semiconductor film were investigated relative to the atomic ratio [14,15,16]. Furthermore, a field-effect mobility of over 10 cm·V^−1^s^−1^ was demonstrated using various dopant materials [17,18,19,20], and a mobility improvement of approximately ten times was achieved by applying chemical treatment or post-treatment [21,22]. In certain cases, the electrical characteristics of the TFT were significantly enhanced via the passivation process [23]. Although the electrical improvements of solution-processed TFTs by atomic composition and doping are closely related to the atomic bonding structure, the effect of atomic doping on crystal structure has not garnered significant attention.

Similar to solution-processed oxide semiconductor TFTs, the electrical performance of plasma-deposited ternary or quaternary oxide semiconductor TFTs is significantly affected by the chemical composition of the oxide semiconductor [24,25,26]. Typically, it has been reported that the crystallinity and conductivity characteristics of the amorphous indium-zinc-gallium-oxide (a-IZGO) semiconductor are significantly affected by the In:Zn:Ga composition ratio. The conductivity characteristics owing to the overlap of the intermetallic hybridization orbital are found to be significantly influenced by the concentration of metal atoms in the oxide semiconductor film. In addition, InO and ZnO have different crystal structures, and the amorphous random network structure is arranged according to the In:Zn:Ga composition ratio. Electrical properties, such as the atomic structure and crystallinity of oxide semiconductors, determine the distribution of tail and trap states in the semiconductor band gap [27,28,29]. These tail and trap states near the oxide semiconductor conduction band (E_C_) are caused by weak bonds and structural defects in the fabrication process. As regards the oxide semiconductor manufactured by plasma sputtering, a thin film with a relatively uniform amorphous random network structure can be fabricated on the basis of the composition ratio of source materials, while the crystal structure of the solution-processed oxide semiconductor is difficult to control.

For solution-processed oxide semiconductor TFTs, the compositional properties of the solution-processed IZO semiconductor are achieved by eliminating the solvent and precursor, and the electrical properties of the fabricated film are different from those of the plasma-vapor-deposited semiconductors [14,15]. In these cases, the morphological and film thickness characteristics play a more decisive role, depending on the crystal structure. Although studies on the solution-processed IZO TFT have focused on the electrical improvement [10,11,12,13] in solution-processed oxide semiconductors, because the atomic bonding structure is rearranged via processes such as solvent evaporation, thermal decomposition, and condensation reactions, the crystallinity is further influenced by the reaction pathways [30]. Therefore, to discuss a one-step forward analysis such as bias-stress mechanisms of TFT, it is essential to understand the trap state distribution and charge transport mechanism based on the atomic bonding structure; however, such an understanding has not been adequately investigated. In addition, the definition of the doping mechanism based on the structural properties can be utilized as a theoretical background for the estimation of the tail, deep, and localized states within the band gap. Hence, the significance of an analytical investigation based on the doping mechanism, with respect to the In molar ratio, should be emphasized for advanced research.

This study entailed an examination of the doping effect of a solution-processed IZO semiconductor with respect to the In molar ratio by using a simplified atomic bonding structure of the component atoms. Furthermore, the formation energy for each bonding structure was classified into an IZO semiconductor band diagram. For this investigation, a solution-processed IZO TFT based on a nitrate precursor was fabricated with respect to the In molar ratio, and its electrical properties were evaluated. The structural and stoichiometric properties of the IZO semiconductor films were examined based on the electrical properties of the IZO TFTs. Regarding the structural characteristics, IZO films were prepared with respect to the In concentration, and the morphology, thickness, crystallinity, and depth profile of the semiconductor films were assessed using scanning electron microscopy (SEM), X-ray diffraction (XRD), and Auger electron spectroscopy (AES) measurements. Moreover, the stoichiometric properties of the IZO films were determined via thermogravimetric analysis (TGA), X-ray photoelectron spectroscopy (XPS), and Hall effect measurements. Consequently, the performance of the IZO TFT was analyzed in relation to the structural/stoichiometric characteristics of the IZO film, and the electron doping effect with respect to the In concentration was examined based on the atomic bonding structure.

## 2. Materials and Methods

The process schematic shown in Figure 1 was used to fabricate a solution-processed IZO TFT with respect to the In molar ratio. Figure 1 shows the fabrication process and chemical reaction pathway for the fabrication of IZO TFT. The research theory in [30] was applied to the chemical reaction pathway of a nitrate precursor-based IZO solution. The IZO solution used in the experiments was prepared by mixing indium nitrate hydrate(In_3_(NO_3_)_3_∙xH_2_O) and zinc nitrate hydrate(Zn_2_(NO_3_)_2_∙xH_2_O) in 2-methoxyethanol(CH_3_O(CH_2_)_2_OH), 2-ME). For vigorous ionization and hydrolysis reactions, the solution was mixed using a magnetic bar at 60 °C for 12 h or more. In this process, the metal-nitrate precursor cluster forms an azeotrope of nitric acid and H_2_O through ligand decomposition, and the bonding structure is simplified. The molar ratio and compositional ratio after the deposition of the IZO solution was determined using the ratio shown in Table 1, wherein the In:Zn molar ratio was empirically determined in the range where the electrical variation of the TFT could be maximized. Note that the 0.25 M of Zn molarity was determined by considering the thickness of the IZO film, and the TFT characteristics with respect to the Zn molar ratio were confirmed in our previous experiments. (refer to [31] for further details).

Before spin-coating of the substrate in Figure 1b, a *p*-type silicon wafer sputtered with a 100-nm-thick SiN_x_ dielectric layer was cleaned with acetone, isopropyl alcohol, and deionized water using an ultrasonicator for 20 min. After drying, the substrate was treated with oxygen plasma for 1 min at approximately 1 mTorr to lower its surface energy. The IZO film was fabricated on the substrate by covering the prepared solution and spin-coating it for 1 min at 2000 rpm. Soft baking was performed for approximately 5 min on a hot plate at 120 °C, under ambient atmospheric conditions. In this process, nitric acid, an azeotrope of H_2_O, and the solvent were evaporated, and the IZO solution was gelated in the form of a metal-nitrate-hydrate bond.

Annealing of the IZO semiconductor film was conducted in a nitrogen atmosphere convection oven at 550 °C for 1 h. For annealing, the speed of heating the substrate was controlled at 5 °C min^−1^, and cooling of the substrate was performed for more than 6 h. In the annealing process of the IZO solution, a nitrogen atmosphere was not required but was used to reduce the external interference. The nitrate precursor in the IZO solution was removed in the form of NO_x_ gas through a metal-nitrate thermal decomposition reaction. The thermal decomposition reactions of In and Zn are initiated at 183 °C and 237 °C, respectively, and residual metal-nitrate bonds are removed at a high temperature of approximately 350 °C or higher. After the thermal decomposition reaction, through a consecutive condensation reaction, the hydroxide bond M-OH is discharged into O_2_ and H_2_O gas to form an O-M-O oxo linkage structure.

On the 20-nm-thick IZO semiconductor film through the annealing process, aluminum source/drain electrodes were thermally deposited through a shadow mask at approximately 1 × 10^−6^ Torr, and the thickness was approximately 50 nm. The IZO TFT was fabricated with a finger-type structure, as shown in Figure 1d. The length and width of the finger-type channel were 2000 (400 × 5) μm and 80 μm, respectively, and W/L = 25. It was assumed that the fringe field effect caused by the un-patterned IZO semiconductor layers was the same for all fabricated devices.

This paper focuses on analyzing the correlation between the electrical properties of IZO TFTs and the fundamental properties of IZO semiconductor films with respect to In concentration. The following equipment and methods were used to measure the electrical properties of the IZO TFT and IZO film. First, the electrical properties of the fabricated IZO TFT were measured using a semiconductor analyzer (ELECS-420) (ELECS, Seoul, Korea) in an ambient lab atmosphere. The presented electrical data were obtained by coordinating more than 200 measurement results of IZO TFTs. The surface characteristics and thicknesses of the IZO semiconductor thin films were measured using field-emission scanning electron microscopy (FE-SEM) (JS-6701F) (JEOL, Tokyo, Japan). Before the SEM surface image measurement, the surface was coated with platinum (≤5 nm) to prevent electron charging on the surface of the semiconductor film. High-resolution X-ray diffraction (HR-XRD) (Rigaku-SmartLab, Tokyo, Japan) was used to analyze the surface and bulk crystal structures of the prepared IZO films. For the analysis, the crystallinity of the surface of the IZO film was measured using the grazing incidence (GI) mode. Furthermore, the depth profile of the IZO film was determined by etching the film from the surface at a speed of 8 nm/min (with reference to SiO_2_) using an electron ion beam. The detection elements for the electron ion beam were C, O, N, Zn, In, and Si, and the measuring instrument was an Auger electron spectroscope (AES) (PHI700) (Ulvac-PHI, Chigasaki, Kanakawa, Japan). Thermogravimetric analysis (TGA) (TGA N-1000) (Scinco, Seoul, Korea) to measure the thermal decomposition temperature of the IZO solution was performed in a nitrogen atmosphere using approximately 30 ± 0.5 mg of IZO solution in a platinum pan, and the IZO solution was heated at a rate of 5 °C min^−1^. For a more detailed analysis of the atomic composition and binding energy of the IZO semiconductor, an x-ray photoelectron spectroscopy (XPS) (K-alpha) (Thermofisher, Waltham, MA, USA) was used. In this analysis, XPS measurements of the IZO film were performed after etching the surface by approximately 5 nm with an electron ion beam to prevent surface contamination. For Hall effect measurement, a solution-processed IZO film of TFT structure is fabricated on a substrate cut to 2 × 2 cm^2^. The carrier concentration, resistivity, and hall mobility of the IZO films with respect to the In molar ratio were analyzed using a Hall effect measurement system (HMS-3000) (Ecopia, Anyang, Korea) by adjusting the input ranges of the current and magnetic field to 1 nA–50 μA and 50 T–0.05 T, respectively.

## 3. Results and Discussion

The electrical performance and operating characteristics of solution-processed IZO TFT were evaluated with respect to the In molar ratio. Figure 2 shows the (a) output characteristics and (b) on-state current values at V_G_ = 40 V and V_D_ = 40 V for IZO TFTs. As shown in Figure 2a, the IZO TFT exhibited typical output characteristics. Except for the on-state current of 0.0125 M, it increased exponentially with respect to the change in In concentration. Furthermore, Figure 3a shows the transfer characteristics with respect to the In molar ratio, and (b) is the sqrt(I_D_) versus gate voltage and threshold voltage versus In molarity graph. Figure 3c shows the representative electrical parameters (on/off ratio, on-state current I_D_on_, field-effect mobility μ_FE_) extracted from the transfer curves, where I_D_on_ is the drain current value when V_D_ = 20 V and V_G_ = 40 V of the transfer curve. The electrical parameter values that can be extracted from Figure 3a–c are summarized in Table 2. The gray box in Figure 3a shows the transfer characteristics of the zinc-oxide (ZnO) TFT without any In composition (0 M). The field-effect mobility shown in Figure 3b was calculated from the maximum slope of the sqrt(I_D_)–V_G_ graph in Figure 3b based on the following equation; advanced field-effect mobility calculation including the mobility in linear region can also be applied for further study [32]:(1)$$I_{D\left(\mathrm{sat}\right)}=\frac{1}{2}\mu_{\mathrm{FE}}C_{\mathrm{ox}}\frac{W}{L}\;{\left(V_{G}-V_{\mathrm{Th}}\right)}^{2},{\;\mu }_{\mathrm{FE}}=\frac{\partial \sqrt{I_{\mathrm{DS}}}}{\partial V_{G}}\sqrt{\frac{2}{C_{\mathrm{ox}}}\frac{W}{L},\;}$$
where I_D(sat)_ is the TFT saturation drain current, μ_FE_ is the field-effect mobility, C_OX_ is the gate capacitance, and W/L is the channel width and length. Threshold voltage V_TH_ was extracted from the sqrt(I_D_)–V_G_ graph, as shown in Figure 3b. As shown in Figure 3a of the transfer characteristics, without the addition of In, the electrical conductivity of the IZO TFT was almost zero. Furthermore, even a small In molar ratio of 0.0125 M resulted in a drastic increase in the on-state current. Thereafter, the on-state current of the transfer curve increased rapidly with respect to the In concentration and the off-state current became uncontrolled above 0.15 M of In ratio; it behaved like a conductor. sqrt(I_D_) also increased with an increasing In molar ratio, and the extracted threshold voltage shifted in the negative direction. As shown in the log-log scale graph in Figure 3c, the on-state current and field-effect mobility were exponentially proportional to the In concentration. As shown in Figure 3 and Table 2, the most ideal TFT operating characteristics with respect to the In molar concentration were observed at 0.125 M; the TFT performance obtained at the fabrication condition of 0.125 M in this study was quite decent, compared to the results in the literature [33,34]. For this TFT, the lower subthreshold voltage swing (S/S) was the lowest, and the on/off ratio was the highest. Although there is a difference in the degree, the electrical characteristics with respect to the increase in the In concentration, that is, the increase in electrical conductivity, the decrease in the threshold voltage, and the increase in the off-state current, can be speculated to be due to the increase in the free carrier density of the IZO semiconductor film. To understand the relationship between the electrical characteristics of the IZO TFT and the structural properties of the IZO semiconductor, SEM images of the fabricated IZO films were measured. Figure 4 shows the surface and vertical images of the IZO film with respect to the In molar ratio. Figure 4a,b are SEM images of a ZnO and indium-oxide (InO) film processed with 0.5 M of Zn and 0.3 M of In molarity, respectively. Figure 4c–f show the SEM images of the IZO film with changes in the molar ratio of In 0.0125, 0.5, 0.125, and 0.2 M, respectively. In contrast to the results for the ZnO film shown in Figure 4a, the thicknesses of the Figure 4b–f films were almost identical at approximately 20 nm. Note that, for a more accurate thickness definition, X-ray reflectometery (XRR) (MFM310) (Rigaku Co., Tokyo, Japan) and high-resolution transmission electron microscopy (HR-TEM) (Spectra 300) (Thermofisher, Waltham, MA, USA) measurements can be applied in future work. Most of the IZO films Figure 4c–f showed aggregation of particles at the top, and Figure 4f showed a tendency to decrease as the In concentration increased. Because there are no aggregated particles in the InO film Figure 4b, it can be suggested that the aggregated particles are ZnO. Note that despite the large electrical changes of the IZO TFT, as confirmed in the output and transfer curves of Figure 2 and Figure 3, there was no significant change in the morphology and thickness of the fabricated IZO film. Nevertheless, the island-like ZnO aggregation can affect electrical properties such as contact resistance and charge injection, so it needs to be analyzed for further investigations through atomic force spectroscopy or SEM—energy dispersive spectroscopy measurement methods. Even with a small amount of In, the electrical performance of the IZO TFT changed significantly, but the surface properties and thickness of the thin films Figure 4c–f were almost the same.

The XRD measurement was performed to analyze the crystallinity of the fabricated 20-nm-thick solution-processed IZO film with respect to the In molar ratio. Figure 5a–c show the XRD analysis results measured in the normal mode for the crystallinity analysis of the IZO bulk film, and Figure 5d–f show the results measured in the grazing incidence (GI) mode for the surface crystallinity of the IZO film. For the measurements, the incident angle ω was set to 0.5° for it not to exceed the critical angle owing to the total external reflection of the IZO film. Furthermore, Figure 5a,d are the measurement results of a ZnO film made of Zn 0.5 M, and Figure 5c,f are the measurement results of an InO film made of In 0.3 M and these are presented for comparison. The peak analyses in the gray box in Figure 5a–c are Si and Si_3_N_4_ lattice characteristic peaks and do not appear in the GI mode because they show surface crystallinity. Several characterized ZnO, IZO, and InO reference crystal peaks are shown at the top of each graph. First, in the crystal properties of the IZO films in Figure 5b,e, the XRD crystallinity was significantly different from that of the bulk and surface. Overall, the properties of the IZO crystal are shown in the bulk, and the crystallinity measured on the surface of the film (GI mode) is close to that of ZnO. As shown in Figure 5b, the IZO bulk film showed IZO crystal properties regardless of the In concentration, and gradually diminished to the InO crystal properties as the In concentration significantly increased (in 0.15 M or more). In the case of the ZnO crystal peaks shown in the result of Figure 5e, it can be explained that the ZnO aggregation layer accumulated on the surface appears in the XPS results. At low In concentrations, ZnO crystal properties were exhibited; however, as the In concentration increased, the InO crystal properties appeared to be highlighted. Thus, it can be estimated that the IZO film has a multilayer structure, the bulk is IZO, and the surface comprises a ZnO aggregation layer. As the In concentration was increased, the entire film exhibited InO crystal properties.

In addition to the structural characteristics of the IZO thin film, its depth profile was measured using AES, as shown in Figure 6. AES measurements of the IZO thin film were performed from the surface to 0–80 nm, and the measurement and detection elements were carbon, oxygen, nitrogen, zinc, indium, and silicon, respectively. In the graph, the 0–20 nm region corresponds to the IZO semiconductor layer, and the shaded region over 20 nm corresponds to the gate dielectric film; the depth profile through XRR or HR-TEM can be considered because it is converted results. The fluctuation characteristics detected on the surface of the IZO film, which are mostly due to C, are caused by contamination at the surface of the ZnO aggregation layer. Here, the In:Zn molar ratio of the IZO thin film presented in the result graph was 0.1 M:0.25 M, and it was similar in measurement results without a notable trend with respect to the In molar ratio, as in the SEM result; Figure 6 presents the most representative example. Similar to the results expected from the XRD results, Zn atoms were mainly detected on the surface of the IZO film, and In atoms were mostly detected at the semiconductor-dielectric interface. Although the electrical properties significantly changed with respect to the In concentration, an apparent trend depending on the In concentration was not observed in the structural properties of the IZO film.

There was no remarkable effect of the In concentration on structural properties, such as roughness, thickness, and crystallinity, and thus, the TGA of the IZO solution was measured to analyze the compositional characteristics of the elements and chemical reaction pathways. The TGA graph measured with respect to In concentration is shown in Figure 7. Figure 7a,b show the organized TGA results according to the temperature range, and (c) shows the weight of the final by-product after the TGA measurement. From the TGA graph, the chemical reaction pathway for IZO solution annealing can be identified in terms of reaction temperature. First, solvent evaporation occurred at approximately 120 °C, and thermal decomposition of the In-nitrate and Zn-nitrate bonds occurred at 184 and 237 °C, respectively. Furthermore, at a higher temperature (approximately 355 °C), residual metal-precursor bonds are released in the gas phase. Finally, the IZO film was fabricated via a condensation reaction by annealing up to 550 °C. A detailed description of the chemical reaction pathway can be found in our previous study [31]. A noteworthy point in the TGA thermal reaction result of the IZO solution with respect to the In concentration is that the weight of the reaction by-product increases, as shown in Figure 7c. The TGA measurements started at almost the same weight as the IZO solution (30 ± 0.5 mg), but the weight of the by-products after the thermal reaction increased linearly with the In concentration. Considering that the thickness of the IZO film is almost the same as the In concentration, it can be determined that the density of the film is significantly changed.

The electrical characteristics of TFT may also be affected by the distribution of oxygen vacancies in semiconductor films. To analyze the stoichiometric and binding energy characteristics of the IZO film with respect to the In concentration, XPS analysis was performed, as shown in Figure 8. XPS peak analysis was measured after 5 nm etching with an electron ion beam from the surface to prevent the influence of surface contamination. Figure 8a,b are graphs of the Zn 2p and In 3d peak binding energy intensities with respect to the In concentration, respectively, and Figure 8c shows the O 1s binding energy peak fitting graph. In the graph of O 1s Gaussian peak analysis, 529.8, 530.0, and 531.0 eV represent metal-oxygen (M-O), oxygen vacancy (Ov), and oxygen-hydroxyl (M-OH) binding energies, respectively. In the O 1s peak analysis result, there was no specific change in oxygen vacancies with respect to the In concentration, the ratio of M-O bonds increased, and the ratio of M-OH bonds decreased. Here, the areal ratio of the metal-oxygen bond gradually increases with an increase in the In concentration, and it can be understood that as the portion of In atoms, which has a lower thermal decomposition temperature than Zn, increases, a higher proportion of O-M-O oxo-link bonds are created. The results are shown in Figure 8a,b, the Zn content of the IZO thin film drastically decreases with respect to the In concentration, and the In concentration increases remarkably. Consequently, it can be noted that even in the case of films with similar thicknesses and crystallinities, the composition ratio of Zn:In atoms can be significantly changed.

The Hall effect was measured to investigate the electrical correlation between the electrical characteristics of the IZO TFT and IZO semiconductor film. The results in Table 3 show the bulk carrier concentration (N_b_), Hall mobility (μ_H_), resistivity (ρ), and sheet resistance (R_s_) measured from the Hall effect characteristics. It can be stated that the Hall mobility μ_H_ is comparable to that of the field-effect mobility μ_FE_ in Table 3, and Both Hall mobility and field-effect mobility were affected by increasing the s-orbital overlap owing to doped In atoms which are relatively larger than Zn atoms, in a ZnO amorphous random network structure. According to the Hall effect measurement results, as the In concentration increased, the resistivity and sheet resistance gradually decreased, and the mobility and bulk carrier concentration effectively increased. This electrical property with respect to the In concentration can be explained by the increase in conductivity owing to the increase in the bulk carrier concentration of the IZO film.

Summarizing the structural properties (roughness, crystallinity, and depth profile) of the IZO film with respect to the In concentration, the IZO film had identical morphology, thickness, and similar crystal properties regardless of the In concentration. In contrast to their similar atomic structural properties, the density and composition ratio were effectively influenced by the In concentration. In addition, the hole mobility and carrier density of the IZO film significantly increased with increasing In concentration. As a result, the change in the electrical characteristics of the IZO TFT with respect to the In concentration in Figure 2 and Figure 3 can be explained by the effect of donor doping of the IZO semiconductor. With an increase in the In concentration, the density of the IZO film increased while having similar crystallinity and thickness in the atomic structure, which is considered to be due to the relative increase in the In atomic composition in the IZO atomic structure. The ZnO structure forms an IZO structure even with a small amount of In. As can be observed from the XPS results, it can be determined that as the In concentration increases, In atoms replace and occupy Zn sites. The donor generated under the influence of In atoms provides free electrons and significantly improves electrical conductivity. It can be speculated that the field-effect mobility of the IZO TFT increases under the influence of the s-orbital overlap of doped In atoms [35,36,37].

Based on the above results, the electrical characteristics of the carrier concentration in the IZO semiconductor film can be theoretically calculated. The estimated major carrier concentrations for In are summarized in Table 4. Based on the saturation current of the IZO TFT, the free electron concentration n of the IZO film can be summarized as follows:(2)$$I_{D\_\mathrm{on}}\propto {q\mu }_{\mathrm{FE}}{\mathrm{nE}}_{D}A,$$
where I_D_on_ is the saturation current of IZO, μ_FE_ is the field-effect mobility extracted from the transfer curve, and Equation (1), where q is the charge amount, E_D_ is the source-drain electric field (V_D_/L), and A is the cross-sectional area of the drain current (AW×d_s_, d_s_ is the thickness of the semiconductor thin film). From the estimation of n in Table 4, the saturation carrier concentration with respect to the In molar ratio can be approximated. Further, the threshold voltage V_Th_ is expressed as follows:(3)$$V_{\mathrm{Th}}=V_{\mathrm{FB}}-\frac{\sqrt{2{q\varepsilon }_{0}\varepsilon_{s}N_{b}2V_{F}}}{C_{i}}+2V_{F},\;V_{\mathrm{FB}}≅\Phi_{\mathrm{MS}}+\frac{Q_{F}}{C_{i}}-\frac{Q_{\mathrm{it}}}{C_{i}}$$
where V_FB_ is the flat-band voltage, ε_0_ is the dielectric constant of the vacuum, ε_s_ is the dielectric constant of the semiconductor, Φ_MS_ is work function difference, Q_F_ is the fixed charge in silicon nitride dielectric, and V_F_ is the Fermi potential; V_F_ = 0.0259 ln (N_b_/n_i_). The IZO semiconductor trap concentration N_T_ was calculated using the relation N_T_ = Q_it_/qd_s_ and the interfacial charge density Q_it_ = qD_it_. In the calculation of the main parameters Q_it_ and N_T_, Q_F_/C_i_ was assumed to be about 30 V; Because the same gate dielectrics were used, it is assumed that the fixed charge value will be similar, and the work function difference is neglected because it is approximately 1 V error. In addition, it was assumed that the charge amount of Q_it_ was uniformly distributed in the ~20 nm of the IZO film. In the analysis of the threshold voltage with respect to the In concentration, the magnitude of Q_s_/C_i_ differs by ~24 V for the 9.0 × 10^14^–8.4 × 10^17^ cm^−3^ bulk concentration N_b_, and the difference in the V_F_ value is approximately 0.3–0.5 V. The calculated Q_it_/q values were 5.7 × 10^11^–5.6 × 10^13^ cm^−2^ as shown in Table 4, and the N_T_ values were 2.8 × 10^17^–2.8 × 10^19^ cm^−3^. Based on the interpretation of Equation (3), the approximately 40 V change in V_Th_ is dominated by a fairly large amount of Q_it_ charge. From the theoretical calculation of V_Th_, it can be observed that the effect of the trap density is more critical than the variation in the bulk doping for the change in the electrical properties with respect to the In concentration. In this study, N_T_ was estimated by simplifying the Q_s_ and Q_it_ values as much as possible; however, the practical N_T_ values were exponentially distributed from the bandgap E_C_. Therefore, for accurate threshold-voltage calculations, [38,39,40] should be used for the distribution of N_T_ as an exponential function as follows:(4)$$\begin{array}{c} V_{\mathrm{Th}}≅V_{\mathrm{FB}}-\frac{\sqrt{2{q\varepsilon }_{0}\varepsilon_{s}}}{C_{i}}[n_{i}\frac{\mathrm{kT}}{q}\exp\frac{{\mathrm{qV}}_{F}}{\mathrm{kT}}\\ \;+N_{T0}E_{T}\exp\left(-\frac{E_{C}}{E_{T}}\right)\left\{E_{T}\left(\exp\frac{q2V_{F}}{E_{T}}-1\right)-2V_{F}\right\}{]}^{1/2}+2V_{F}, \end{array}$$
where the n_i_ is the intrinsic carrier density of the semiconductor, and kT is 0.0259 eV. Assuming that the trap density distribution decreases exponentially from E_C_, N_T0_ is the trap density at E_C_ and E_T_ is the characteristic energy that is related to the slope; where the E_C_ was 1.6, which is half of the band gap and V_FB_ in Equation (4) is V_FB_ = Φ_MS_ + Q_F_/C_i_ by replacing the effect of interfacial charge in N_T_. According to the calculation of Equation (4), the trap density at E_C_ was distributed as 10^19^ cm^−3^ to 10^22^ cm^−3^ from E_C_ for an In concentration of 0.0125 M to 0.2 M of IZO TFTs, and the characterization energy was 0.46 to 0.04 eV. A detailed distribution of acceptor-like trap states within the IZO band gap can be estimated through thermal energy analysis and will be introduced through our improved study in our future work. Additionally, in Figure 3, it can be assumed that the off-state current, which increases with the concentration of In, is the effect of the increase in the gate-source/drain reverse recombination-generation current as follows [41,42]:(5)$${\;J}_{R}\propto {\mathrm{qWN}}_{T}e_{p}$$
where J_R_ is the leakage current density between the gate-to-drain electrode and W and e_p_ are the depletion width and emission rate constant, respectively. As the number of traps increases, the number of recombination-generation sites increases, and the leakage current increases.

The following results were inferred based on the structural/stoichiometric properties of the IZO semiconductor and the electrical properties of the IZO TFT. The role of ZnO in the solution-processed IZO semiconductor film is presumed to form an amorphous random network structure during the fabrication process. In addition, it can be determined that the role of In atoms in the solution process is to increase the free carrier density of the IZO semiconductor by structurally replacing Zn atoms. From the analysis results of the threshold voltage, it was speculated that these free carriers were primarily influenced by the trap charge. Based on the observed results, a simplified atomic model for In doping is shown in Figure 9. Based on the XRD crystal structure analysis results, the solution-processed ZnO film had an amorphous random network structure, as shown in Figure 9a. In the case of such amorphous materials, the trap state is caused by structural defects. Moreover, in IZO semiconductors, contrary to crystalline Si (c-Si), the number of bonds is determined by the bonding coordination, and the number of coordination is determined according to Mott’s “N,8-N bonding rule” [42,43,44]. In addition, defect bonding in an amorphous random network requires a valence bond as the defect. As shown in Figure 9b, the stable Zn-O bond maintains its bonding structure through an ionic bond with two surrounding oxygen atoms and forms oxygen vacancies owing to the structural defects in the amorphous random network. Conversely, the In-O bond maintains a stable structure through an ionic bond with three oxygen atoms, and when oxygen atoms are insufficient, acceptor-like states are formed, as shown in Figure 9c, a state equivalent to a dangling bond in a single crystal structure with covalent bonding. In Figure 9d, weak bond breakage among the ionic bonds of the ZnO atomic bond is either an acceptor-like state D_ZnO_ (D_ZnO_^−^ + V_O_^−^) or a donor-like state of oxygen vacancy V_O_ (V_O_^+^ + e^−^) can be created depending on the energy position. Additional reactions, such as charge trapping and detrapping, occurred by V_O_ and D_ZnO_ in this model. However, because the electrical conductivity of the ZnO TFT without In was extremely low; it was estimated that the concentration of this bonding structure was relatively low. Furthermore, the In-doping model shown in Figure 9e explains that the In atom in the basic ZnO structure in Figure 9d is replaced by Zn atoms. By replacing the two-fold coordinated Zn with a three-fold coordinated In dopant, an ionized In dopant and a negatively charged InO acceptor-like state D_InO_^−^ are generated to sustain the net equilibrium state. A considerable amount of the acceptor-like trap D_InO_^−^ is ionized and electrons (D_InO_^−^→D_InO_^0^ + e^−^) become free electrons, forming a neutral state as a whole. As the In concentration increased, the Zn atoms in the ZnO structure were replaced by In atoms, thereby increasing the IZO film density and increasing the total concentration of free electrons.

Considering the above results, Figure 10 shows the hybridization mechanism and formation energy of the IZO structure. As shown in Figure 10, the valence electron energy of a metal oxide material is arranged as in Figure 10a in a vacuum state. Furthermore, the two-/three-coordinated Zn 4s/In 5p and In 5s orbitals are the same as in Figure 10b. In this case, the hybridization orbital rearranged by the Madelung potential can be expressed as shown in Figure 10c,d. Note that owing to weak bonding and structural defects, additional energy states, such as the tail state and trap state (D_ZnO_, D_InO_) can be positioned inside the band gap. Finally, Figure 10e shows the formation energy of the entire IZO semiconductor, in which the formation energies of Figure 10c,d are superimposed. The position of the conduction band minimum (E_C_) and the valence band maximum (E_V_) of the IZO semiconductor were defined by considering the In 5s and O 2p orbitals, and these energy positions were sketched based on [2,35]. The formation energies of the In-O and Zn-O defects exist in the shallow and deep states near the E_C_ of the band gap, as shown in Figure 10e. Because the electrical properties of IZO TFTs are significantly improved by the influence of In concentration, the location of the In-O defects will be close to the E_C_, and the Zn-O defects will be relatively deep and have low concentrations. In addition, note that the distribution of acceptor-like states inside the bandgap is determined not only by the defect traps but also by the weak bond between the Zn-O and In-O structures in Figure 9; the tail state due to the weak bond is hidden in Figure 10. It was assumed that the formation energy of this tail state was almost the same as the DOS distribution of the acceptor-like state. Based on the schematic shown in Figure 10e, the following conclusions can be drawn: As the In molarity increases, the formation energy D_Zn_^0^ of Zn-O is replaced by the In-O formation energy D_In_^−^, the subsequent density of the shallow state increases, and the density of the deep state decreases. Therefore, the E_F_ level of the IZO semiconductor shifts towards the conduction band E_C_ [45,46]. Although this model was used as an example to explain the In-doping process in a solution-processed IZO semiconductor, a similar theory can be applied to other ternary/quaternary element oxide semiconductors. The proposed model in this study is also expected to be utilized in further studies on the calculation of the band gap state distribution and the analysis of the bias stress effect of the solution-processed IZO semiconductor.

## 4. Conclusions

In this study, we suggested the simplified atomic bonding structure and In doping model in the energy band diagram for solution-processed IZO semiconductors. For this investigation, the structural characteristics of the IZO semiconductor and the electrical characteristics of the IZO TFT were analyzed with respect to the In molar ratio. The optimal characteristics of the IZO TFT were obtained at the composition condition of the In:Zn ratio of 14.4:16:3, which were found to be comparable to the results reported in the literature. Moreover, the electrical conductivity of the IZO TFT increased significantly with increasing In concentration, and the threshold voltage shifted in the negative direction. The structural characteristics of the IZO film were analyzed by SEM, XRD, and AES, and no specific trends with respect to the In concentration were observed in the IZO films. The fabricated IZO thin film had a multilayer structure; the bottom layer was IZO, and the surface was formed of a ZnO aggregation layer. Although the IZO films did not show significant structural changes with respect to In concentration, the stoichiometric properties of the IZO films changed significantly. These changes were demonstrated through TGA, XPS, and Hall effect measurements. The structural density and In composition ratio of the films significantly increased with respect to the In molar ratio, and the subsequent carrier concentration of the bulk film remarkably increased. Herein, despite the structural similarity of the IZO film, the enhancement in the electrical properties of the IZO TFT can be explained by the increase in the carrier concentration with respect to the In concentration. Consequently, it is reasonable to conclude that the donor doping effect by these carriers contributes to an increase in the acceptor-like states owing to structural defects, which was proposed as a model for the atomic bonding structure and trap state of IZO semiconductors in this study. We believe that analyses of the bonding structure between atoms and the energy structure thereof are essential to understand the change in electrical properties according to the material composition of the solution-processed oxide semiconductor. This study can be used in future studies to improve the performance of solution-processed oxide TFTs by combining different material compositions and doping techniques.

## Figures and Tables

**Figure 1 materials-15-06763-f001:**
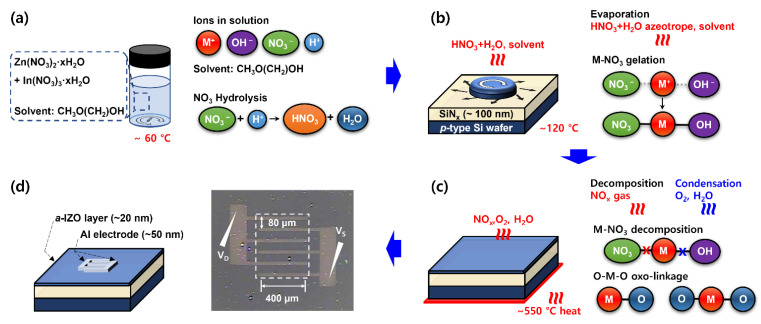
Process schematic and chemical reaction pathway used to fabricate IZO TFT. (**a**) IZO solution formulation, (**b**) spin-coating process, (**c**) annealing process, and (**d**) thermal deposition process and TFT illustration.

**Figure 2 materials-15-06763-f002:**
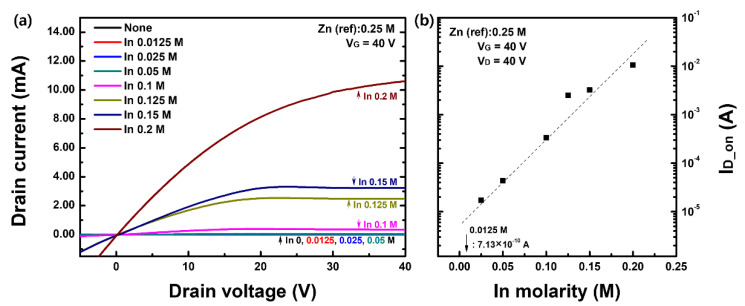
(**a**) Output characteristics and (**b**) on-state current of IZO TFT with respect to the In molar ratio.

**Figure 3 materials-15-06763-f003:**
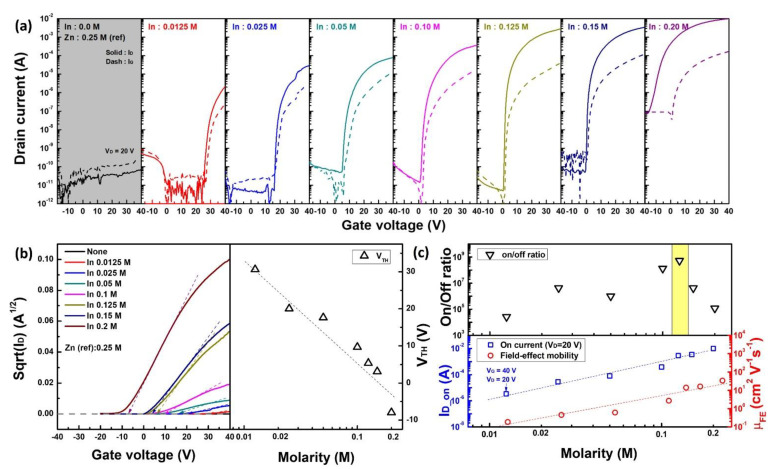
Major electrical properties of solution-processed IZO TFT with respect to the In molar ratio. (**a**) Transfer curve characteristics, (**b**) square-root drain current graph for threshold voltage extraction, (**c**) significant electrical parameters.

**Figure 4 materials-15-06763-f004:**
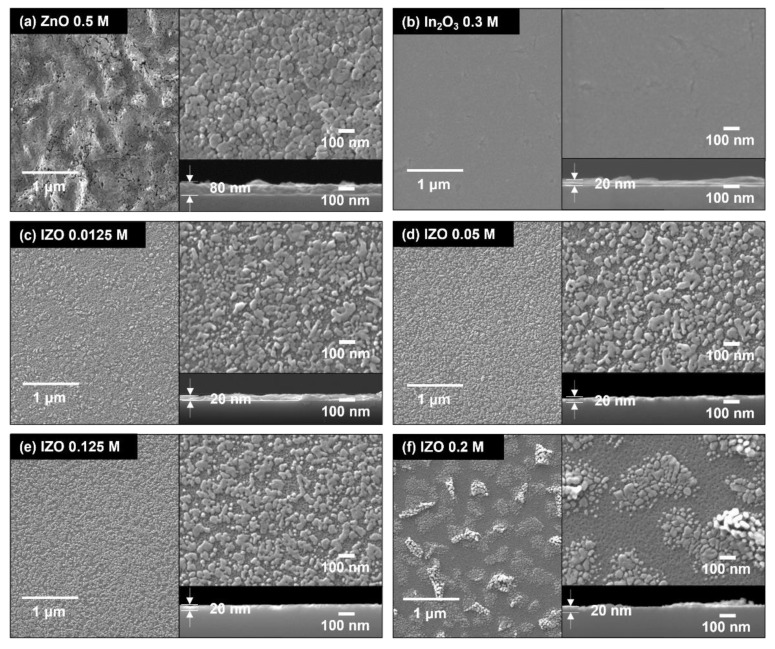
SEM images of IZO TFT with respect to the In molar ratio. For comparison, (**a**) 0.5 M of ZnO film image and (**b**) 0.3 M of InO film image are attached, respectively. The reference Zn molar of (**c**–**f**) was fixed at 0.25 M.

**Figure 5 materials-15-06763-f005:**
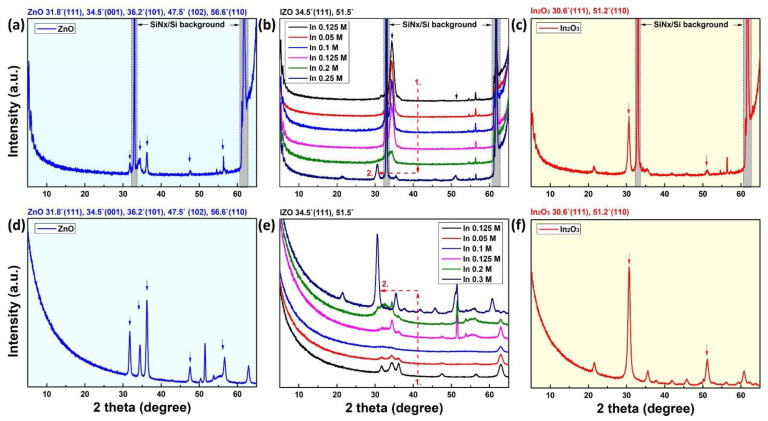
XRD measurement graphs for crystallinity characteristic of solution-processed IZO films. (**a**–**c**) are normal mode results for bulk crystallinity, (**d**–**f**) are grazing incidence results for surface crystallinity. (**a**,**d**) and (**c**,**f**) are the results of ZnO and InO film for reference, respectively.

**Figure 6 materials-15-06763-f006:**
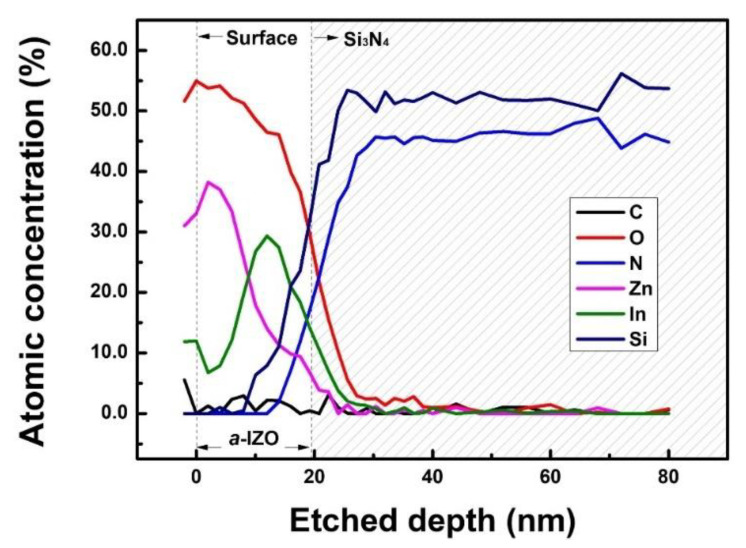
AES results of solution-processed IZO film. The In molarity of IZO film was 0.1 M.

**Figure 7 materials-15-06763-f007:**
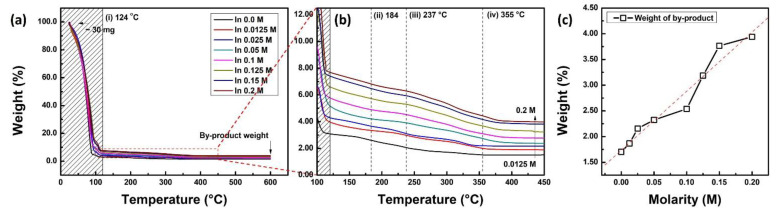
(**a**) The weight loss graph of IZO solution with respect to In molar ratio in the entire annealing region. (**b**) Enlarged TGA graph in thermal decomposition region. (**c**) % weight of final by-product of IZO solution with respect to In molar ratio.

**Figure 8 materials-15-06763-f008:**
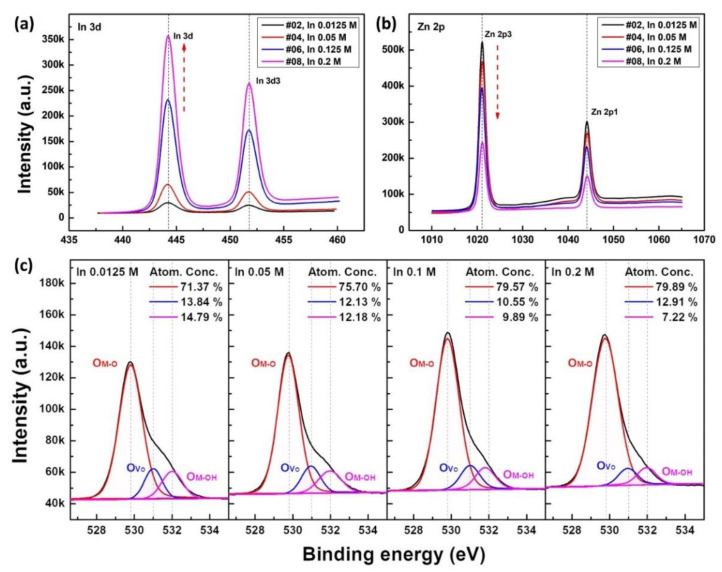
XPS peak analysis of solution-processed IZO film with respect to the In molar ratio. (**a**–**c**) are the analysis results of In 3d peak, Zn 2p peak, and O 1s peak results, respectively.

**Figure 9 materials-15-06763-f009:**
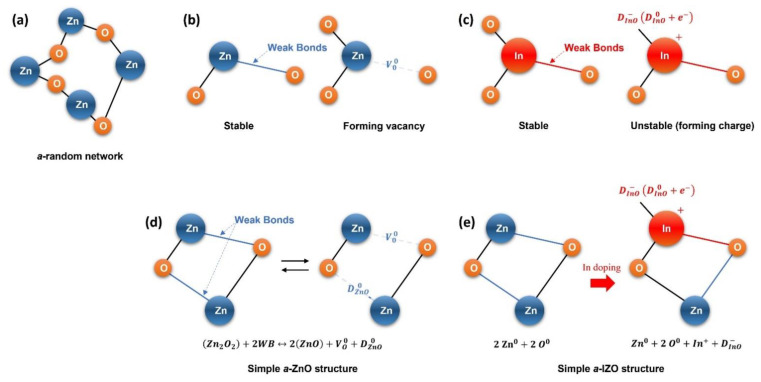
Schematic of simple atomic bonding model of IZO semiconductor. (**a**) Zn-O amorphous random network, (**b**) Zn-O boning, and (**c**) In-O bonding. (**d**) shows the weak bond breaking in Zn-O bonding structure, and (**e**) represents the replacement mechanism of Zn atom by In doping.

**Figure 10 materials-15-06763-f010:**
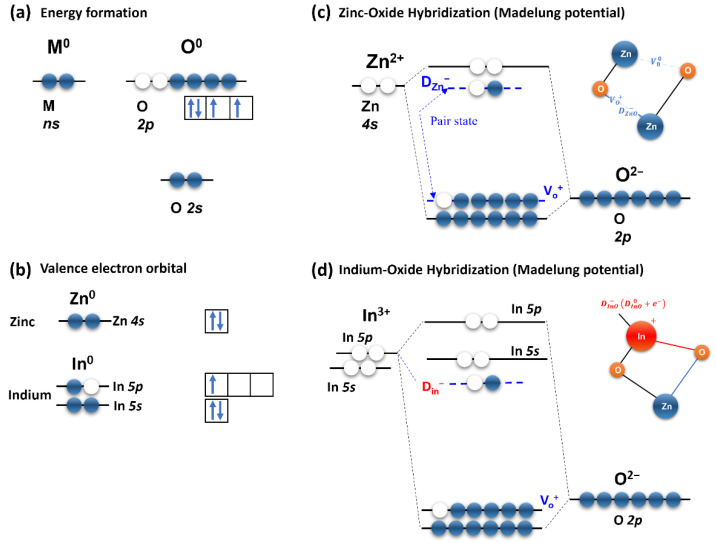
Imaginary band gap energy formation diagram and hybridization mechanism of IZO semiconductor.

**Table 1 materials-15-06763-t001:** Molarity table of IZO solution for IZO TFT fabrication (the Zn molarity of solution is fixed).

No.	In01	In02	In03	In04	In05	In06	In07	In08
In molarity (M)	None	0.0125	0.025	0.05	0.1	0.125	0.15	0.2
Zn molarity (M)	0.25 M (fixed)							
In:Zn ratio in the film	0	1.4:16.3	2.9:16.3	5.7:16.3	11.5:16.3	14.4:16.3	17.2:16.3	22.3:16.3

**Table 2 materials-15-06763-t002:** Representative electrical parameters of solution-processed IZO TFT with respect to the In molar ratio.

Molarity (M)		Mobility (cm^2^·V^−1^ s^−1^)	I_on_ (V_G_ = 40 V) (A)	On/Off Ratio	V_TH_ (V)	S/S (V/decade)
In	Zn					
0.0	0.25 (fixed)	Inactive				
0.0125		0.1859	3.35 × 10^−6^	~2.6 × 10^5^	30.61	2.58
0.025		0.4497	2.85 × 10^−5^	~4.2 × 10^6^	20.03	1.99
0.05		0.6084	7.97 × 10^−5^	~1.0 × 10^6^	17.63	1.65
0.1		2.704	3.85 × 10^−4^	~1.3 × 10^8^	9.64	1.68
0.125		13.90	2.76 × 10^−3^	~5.3 × 10^8^	5.35	0.98
0.15		16.03	3.41 × 10^−3^	~4.2 × 10^6^	3.10	0.93
0.2		34.18	9.88 × 10^−3^	~1.2 × 10^4^	−7.95	4.992

**Table 3 materials-15-06763-t003:** Hall effect measurement results of solution-processed IZO semiconductor with respect to the In molar ratio. For Hall effect measurement of 1 to 8 samples, the input current was 1 × 10^−9^, 50 × 10^−9^, 200 × 10^−9^, 1 × 10^−6^, 200 × 10^−9^, 10 × 10^−6^, 50 × 10^−6^, and 50 × 10^−6^ A, respectively. The input magnetic field was 50, 10, 5, 0.05, 0.5, 0.2, 0.1, and 0.05 T, respectively.

Sample No.	In Molarity [M]	N_b_ [#·cm^−3^]	µ_H_ [cm^2^·V^−1^s^−1^]	R_s_ [Ω/sqr]	ρ [Ω·cm]
1	0	1.495 × 10^15^	0.4898	4.263 × 10^9^	8.526 × 10^3^
2	0.0125	9.052 × 10^14^	1.368	2.521 × 10^9^	5.042 × 10^3^
3	0.025	1.423 × 10^16^	4.130	5.311 × 10^7^	1.062 × 10^2^
4	0.05	7.998 × 10^17^	7.762	5.027 × 10^5^	1.005 × 10^2^
5	0.1	1.121 × 10^18^	14.69	1.896 × 10^5^	3.792 × 10^−1^
6	0.125	1.516 × 10^18^	19.99	1.030 × 10^5^	2.060 × 10^−1^
7	0.15	5.737 × 10^18^	23.86	2.280 × 10^4^	4.560 × 10^−2^
8	0.2	8.368 × 10^18^	35.79	1.042 × 10^4^	2.085 × 10^−2^

**Table 4 materials-15-06763-t004:** Estimation of some representative carrier concentrations with respect to the In molar ratio.

Molarity [M]		N_b_ [#·cm^−3^]	n(sat) [#·cm^−3^]	D_it_ [#·cm^−2^]	N_T_ [#·cm^−3^]
In	Zn	(Measured)	(Estimated)		
0	0.25 (Fixed)	1.495 × 10^15^	Inactive	Inactive	Inactive
0.0125		9.052 × 10^14^	1.125 × 10^17^	−5.710 × 10^11^	−2.855 × 10^17^
0.025		1.423 × 10^16^	3.956 × 10^17^	3.435 × 10^12^	1.717 × 10^18^
0.05		7.998 × 10^17^	8.888 × 10^17^	1.774 × 10^12^	8.869 × 10^17^
0.1		1.121 × 10^18^	1.239 × 10^18^	4.473 × 10^12^	2.236 × 10^18^
0.125		1.516 × 10^18^	1.239 × 10^18^	5.619 × 10^12^	2.810 × 10^18^
0.15		5.737 × 10^18^	1.306 × 10^18^	2.399 × 10^12^	1.199 × 10^18^
0.2		8.368 × 10^18^	1.804 × 10^18^	5.123 × 10^12^	2.562 × 10^18^

## Data Availability

The research data presented in this study are available on request from the corresponding author.

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
