# Peer review of "Investigation on Atomic Bonding Structure of Solution-Processed Indium-Zinc-Oxide Semiconductors According to Doped Indium Content and Its Effects on the Transistor Performance"

_materials, 2022, doi:10.3390/ma15196763_

Round 1

Reviewer 1 Report

In this manuscript, the authors reported the relationship between the film properties of nitrate precursor-based amorphous indium-zinc-oxide (a-IZO) semiconductors and the electrical performance of solution-processed a-IZO TFTs with respect to the In molarity ratio. By increasing the In molarity ratio, the carrier concentration increased. As a result, the electrical properties of the a-IZO TFT improved. In terms of device performance, it looks very nice. However, I strongly recommend authors clarify several issues. Detailed comments are:

1.       The influence of the In/Zn ratio of IZO films on film properties and TFT characteristics has already been reported. For example, the following paper (https://doi.org/10.1149/1.3298886).  What is the novelty of your paper? In the introduction, author should mention more in-depth the previous studies on solution-processed IZO films and TFTs.

2.       Author has mentioned the In/Zn ratio of the solution, but the In/Zn ratio of the IZO film after deposition should also be mentioned.

3.       According to the SEM image, IZO appears to be island-like and there seems to be a large variation in roughness and composition. Thus, the surface roughness by AFM and composition analysis by EDS should be performed.

4.       The author has determined the IZO film thickness (20 nm) by SEM images, but other methods should be used to determine the film thickness more accurately.

5.       IZO Film density should be quantitatively determined by XRR. Film thickness determined by SEM is not reliable.

6.       The XRD results show diffraction peaks, but why do you mention amorphous IZO in the manuscript?

7.       The mobility of the IZO film obtained by Hall effect measurements shows 365.3-7040 cm2V-1s-1, which is too high for IZO and not reliable. Even single-crystal In2O3 films have a Hall mobility of about 160-200 cm2V-1s-1.

8.       What is the origin of the carriers generated by the increase in the In molar ratio? From the XPS results, there is no change in the amount of oxygen vacancies in the IZO film.

9.       Why does the gate leakage current of IZO TFTs increase with increasing In mole ratio?

10.   The horizontal axis in Figure 2(a) is the drain voltage.

11.   Units are required for Nb and trap density.

Author Response

To

The Reviewer

MDPI Materials

RE: Author’s replay to the review report (Reviewer 1)

Dear Sir/Madamn:

First of all, we sincerely appreciate your professional comments and truly thank you for reading our paper thoroughly, as well. It was a such pleasure to have a novel conversation with an expert in this research area. Your suggestions have promoted us to significantly consider this topic. As you mentioned, some parts we made were wrong, and some descriptions need to be corrected. We look forward to having an in-depth discussion on our research topics to be published in near future, too.

And in your considerate comments, we have explained and made modifications as follows:

  1. Understanding the atomic bonding structure and doping mechanism of solution-processed IZO TFT is important to accurate modeling/simulating the properties of semiconductor. This work can be interesting because this paper tried to explain the electrical properties by the increase in bulk carriers, which has not been dealt in other paper

As Reviewer mentioned, we have included the similar researches in this field [https://doi.org/10.1149/1.3298886, https://iopscience.iop.org/article/10.1149/1.2969451, https://doi:10.1016/j.tsf.2009.01.151 https://doi.org/10.1063/1.3151827] and modified the introduction to clearly convey our intentions.

** To add our thoughts, the fundamental configuration of doping mechanism is of prime importance for characterizing the density of state distribution. Based on the fundamental definitions of this paper, calculating/simulating the band gap state can be properly performed and developed. Figure R1 introduces the investigation on the DOS calculation and extraction of AOS semiconductor with respect to the In molarity ratio. Certain conclusions of this paper were speculated based on the study in Figure R1 and will be utilized as fundamental features for the paper to be published. The definition of the atomic bonding structure and doping mechanism for DOS calculation is special compared to studies in this research area and the fundamental features of this work can be extended to research such as Figure R1.

*Figure R1 is included in the PDF file we attached.

  1. As Reviewer mentioned, the compositional property after the deposition is critical because the In content will be a dominant effect on the electrical behavior of TFT.

The composition ratio after the annealing process is calculated and attached in Table 1. And a description added as composition ratio is added. 

  1. We agreed that the morphological characteristics and thickness of the semiconductor film show a dominant effect on electrical characteristics on IZO TFT. Thus, the roughness and the compositional characteristics of surface of the film should be investigated through AFM and SEM-EDS. In this paper, we have focused on AES and XPS measurements because we have speculated that the electrical properties of IZO TFT are noticeably affected by the In composition ratio inside the semiconductor. So, the effect of the surface was only referred to as GI-XRD and AES.

Nevertheless, the island-like ZnO aggregation can damage electrical properties such as contact resistance and charge injection, we added the necessity of AFM and SEM-EDS measurement for further investigation in the manuscript.   

  1. As Reviewer commented, investigation of thickness features is of paramount importance, especially for those who dealing with calculations and simulations. We presumed that the thickness of the IZO film through the measurement results such as SEM, alpha-step, and depth profile, speculated based on our empirical conclusion that the thickness is constant regardless of fairly large amount (0.4 M) of the In concentration; rather, it was highly dependent on the Zn concentration.

Still, we agree that Reviewer’s comments on the measurement methods of thickness will significantly improve the data accuracy. Therefore, the necessity of thickness profile through XRR and HR-TEM measurement was attached in the manuscript.

  1. As stated, the data accuracy of the thickness will significantly improve the results of this paper. The necessity of measurement results such as XRR and HR-TEM is added in the manuscript.

  1. It is absolute true that the ZnO and InO have clear crystallinity and we have proved that XRD results in Figure 5. Thus, the Reviewer’s comment needs to be reflected in the revised manuscript.

As Reviewer’s comments, In this paragraph and the descriptive expression related crystallinity, the word ‘amorphous’ has been modified or deleted to avoid misleading.

  1. As more than 200 electrical data were analyzed, in the case of Hall measurement, the data was extracted by averaging more than 10 results. In this process, it was confirmed that repeated measurements caused serious errors in the data, and extremely high magnitude that should not be included were calculated in average.

A serious error in Hall mobility were corrected, and the results of Hall measurement were re-examined thoroughly.

  1. We have fully agreed with the fact that the oxygen vacancy is the major reason for the electrical enhancement of TFT performance.

However, as Reviewer mentioned, there is no change detected in the oxygen vacancy with respect to In molarity ratio. We have tried to explain the invariant value of oxygen vacancy is correlated with the fixed Zn molarity ratio. The basis of an amorphous random network is formed by the ZnO composition of IZO solution, and the carriers are generated by replacing the Zn atoms with In atoms in the amorphous random network structure. We have explained that ionized electrons through the unbounded bond caused by replacing Zn atoms are the origin of carriers. [Check: p.14 lines 431-437]

This explanation on carrier generation is highlighted in the manuscript.

  1. In off-state region, the transistor is biased about 40 V in the gate (-20 V) to drain (20 V). This can be presumed the forward biased p-i-n (p-silicon-gate insulator-IZO semiconductor) diode, increasing the bulk carrier density by In concentration can be regarded as the increase of ‘n’.

As Reviewer pointed out, it is stated in the manuscript that this leakage current is an off-state current. We skipped the equation for a brief explanation, however, re-attatched to avoid any misleading.   

  1. Thanks for pointing out. The x-axis of output current graph has been revised. 

  1. The units for the bulk carrier density, charge trap density, interfacial trap density are added in cm-3, cm-3, and cm-2, respectively.

We believe that it would have been a much better study if your suggestion had been properly taken into account.

Thanking you in anticipation,

With Best Wishes,

Dr. Prof. Jaehoon Park.

Department of Electronical Engineering

Hallym University

Tel : +82-33-248-2357

e-mail : [email protected]

Reviewer 2 Report

This is one of the well-written manuscripts I have read recently in the field. It is an interesting work and I am satisfied with the discussion and conclusion. The text is well written, clear and easy to read. The conclusions are consistent with the evidence and arguments presented. And the discussion part is the most interesting portion of the work.

This works reports on variation in the electrical properties of IZO thin film transistors and the atomic bonding structure was evaluated using AES, XPS, and hall measurements. It is relevant to the scope of the journal and interesting work. When compared to other similar works, this study not only reports the properties of IZO thin films and TFTs but also, explains the electronic properties of the fabricated IZO thin films.

However, since the calculation of mobility and threshold voltage played a vital role in deriving the conclusion, I strongly recommend reading the following paper before making the calculations.

Choi, H. H. et al., Critical assessment of charge mobility extraction in FETs. Nat. Mater. 171(17), 2–7 (2017).

Also, I prefer to change the title as it has tested the TFTs and the bonding structure study needs further pieces of evidence for making the conclusion.

Author Response

To

The Reviewer

MDPI Materials

RE: Author’s replay to the review report (Reviewer 2)

Dear Sir/Madamn:

First of all, we sincerely appreciate your professional comments and truly thank you for reading our paper thoroughly, as well. We believe your suggestions have promoted us to significantly consider this topic. We look forward to having an in-depth discussion on our research topics to be published in near future, too.

And in your considerate comments, we have explained and made modifications as follows:

  1. The title of this paper has been slightly revised to reflect the Reviewer’s opinion.

  1. As Reviewer mentioned, we have considered the several calculating methods for field-effect mobility extraction. The field-effect mobility in linear region can be effectively applied to our electrical data. However, in some samples, the linear mobility was not defined because the channel was not properly induced at relatively low gate voltage (1~2 V).

Although these data were not included in the paper, we agreed that these calculating methods can significantly enhance the data accuracy and introduced in the manuscript.

We believe that it would have been a much better study if your suggestion had been properly taken into account.

Thanking you in anticipation,

With Best Wishes,

Dr. Prof. Jaehoon Park.

Department of Electronical Engineering

Hallym University

Tel : +82-33-248-2357

e-mail : [email protected]

Round 2

Reviewer 1 Report

The authors have made significant revisions in their resubmitted paper, but there are still some serious problems. I strongly recommend authors clarify several issues. Detailed comments are:

1.     The Hall effect measurement results have been corrected in the revised manuscript, but the data are unreliable. In general, resistivity (ρ) is defined by the following equation; ρ = 1/(qNbμ). q, Nb and μ are the elementary charge, carrier concentration and mobility, respectively. For example, Sample No.1 in Table 3, Nb = 1.704×1017 cm-3, μ = 4.43 cm2V-1s-1, and thus ρ should be 8.28 Ω・cm. However, ρ shows 2.42×10-3 Ω・cm.

2.     The standard deviation for Hall measurement should be added because it looks very large variability.

3.     Why does the field-effect mobility increase with increasing In ratio in TFTs but not in Hall measurements?

Author Response

To

The Reviewer

MDPI Materials

RE: Author’s replay to the review report (Reviewer 1, round 2)

Dear Sir/Madamn:

Detailed response on the Reviewer's comment is attatched to the 'replay to reivew report_1 v2.0.pdf' file. 

Thanking you for reviewing our paper and in anticipation,

With Best Wishes,

Dr. Prof. Jaehoon Park.

Department of Electronical Engineering

Hallym University

Tel : +82-33-248-2357

e-mail : [email protected]
